# Regulation of the Activity of the Dual Leucine Zipper Kinase by Distinct Mechanisms

**DOI:** 10.3390/cells13040333

**Published:** 2024-02-11

**Authors:** Kyra-Alexandra Köster, Marten Dethlefs, Jorge Duque Escobar, Elke Oetjen

**Affiliations:** 1Department of Clinical Pharmacology and Toxicology, University Medical Centre Hamburg-Eppendorf, 20246 Hamburg, Germany; koester.kyra@gmail.com (K.-A.K.); m.dethlefs.ext@uke.de (M.D.); 2DZHK Standort Hamburg, Kiel, Lübeck, Germany; j.duque-escobar@uke.de; 3University Center of Cardiovascular Science, Department of Cardiology, University Heart & Vascular Center Hamburg, University Medical Center Hamburg-Eppendorf, 20246 Hamburg, Germany; 4Institute of Pharmacy, University of Hamburg, 20146 Hamburg, Germany

**Keywords:** dual leucine zipper kinase, phosphorylation, protein–protein interaction, proteasomal degradation, palmitoylation

## Abstract

The dual leucine zipper kinase (DLK) alias mitogen-activated protein 3 kinase 12 (MAP3K12) has gained much attention in recent years. DLK belongs to the mixed lineage kinases, characterized by homology to serine/threonine and tyrosine kinase, but exerts serine/threonine kinase activity. DLK has been implicated in many diseases, including several neurodegenerative diseases, glaucoma, and diabetes mellitus. As a MAP3K, it is generally assumed that DLK becomes phosphorylated and activated by upstream signals and phosphorylates and activates itself, the downstream serine/threonine MAP2K, and, ultimately, MAPK. In addition, other mechanisms such as protein–protein interactions, proteasomal degradation, dephosphorylation by various phosphatases, palmitoylation, and subcellular localization have been shown to be involved in the regulation of DLK activity or its fine-tuning. In the present review, the diverse mechanisms regulating DLK activity will be summarized to provide better insights into DLK action and, possibly, new targets to modulate DLK function.

## 1. (Patho)physiological Actions of DLK

The dual leucine zipper kinase (DLK) alias mitogen-activated protein 3 kinase 12 (MAP3K12) has gained much attention in recent years due to its involvement in several neurodegenerative diseases such as amyotrophic lateral sclerosis (ALS), Parkinson’s disease and Alzheimer’s disease [1,2,3,4,5,6], and glaucoma [7] (Table 1). In an insulin-producing beta-cell line, DLK was shown to become activated by prodiabetogenic signals, inhibit insulin synthesis and secretion, and induce apoptosis, thus promoting the pathogenesis of diabetes mellitus type 2 [8,9,10,11,12,13]. Furthermore, in the next-generation sequencing data set GSE81608, the upregulation of *MAP3K12* was observed in the islets of type 2 diabetic patients in comparison to healthy donors [14], and in a genome-wide association study, the SNV rs77511173-C (location 12.53489753) within *MAP3K12* was associated with body mass index in a Japanese population [15]. However, in a rat model, DLK was required for postnatal beta-cell proliferation [16]. Thus, DLK might be important for the development of the neuronal system [17] and early postnatal beta-cells [16].

**Table 1 cells-13-00333-t001:** Involvement of DLK in neurological diseases.

Modeled Disease or Disease Mechanism	In Vivo/In Vitro Model (Organism)	Experimental Disease Model	Involvement of DLK	Ref
Seizure/excitotoxic neurodegeneration	Mouse	Kainic acid	Inducible DLK knock-out reduced neuronal degeneration.	[18]
Amyotrophic lateral sclerosis (ALS)	Mouse	SOD1^G93A^ mutation	Deletion of DLK preserved large myelinated axons and protected neurons in lumbar spinal cord.	[19]
	Mouse	SOD1^G93A^ mutation	Dual inhibition of DLK and LZK suppressed elevation of p-c-Jun and neurofilament light (Nf-L).	[20]
	Mouse Mouse primary neurons Human tissue (postmortem)	SOD1^G93A^ TDP-43^A315T^ Trophic factor deprivation ALS patients’ lumbar spinal cord	Deletion or inhibition of DLK reduced p-c-Jun; protected against axon degeneration, neuronal loss, and functional decline; and improved survival of SOD1^G93A^ mice. p-JNK and p-c-Jun were elevated in patient lumbar spinal cord lysates.	[1]
Alzheimer’s disease (AD)	Mouse Human tissue (postmortem)	PS2APP Tau^P301L^ AD patients’ brain	Deletion of DLK reduced p-c-Jun in immunoblot and immunochemistry and improved preserved cognitive function. P-c-Jun was detected in dentate gyrus via immunohistochemistry.	[1]
	Human ESC-derived cortical neurons Mouse	ApoE2, ApoE3, ApoE4 treatment	ApoE increased DLK, while ApoE3 slowed DLK protein turnover rate.	[2]
	Mouse primary hippocampal and cortical neurons	Overexpression of human tau	DLK inhibition reduced p-JNK and p-c-Jun and cytotoxicity	[6,21]
	Mouse	APP/PS1 mouse model	Downregulation of DLK by MiR-191-5p inhibited Amyloid-beta-induced microglial cell injury.	[22]
Parkinson’s disease (PD)	Mouse	6OHDA (neurotoxin) (intra-striatal)	AAV-directed expression of d/n-DLK (K152A) had anti-apoptotic and trophic effects on dopaminergic neurons.	[23]
	Mouse	MPTP (mitochondrial toxin)	DLK inhibition suppressed c-Jun phosphorylation in dopaminergic neurons of the substantia nigra.	[24]
Traumatic brain injury	Mouse	Impact acceleration	Injury activated the DLK-JNK axis in retinas, while DLK deletion blunted c-Jun-phosphorylation.	[25]
Traumatic brain injury (subarachnoid hemorrhage)	Rat	Endovascular perforation	DLK silencing attenuated brain edema and neuronal apoptosis and improved neurobehavioral functions. DLK overexpression deteriorated neurobehavioral functions and brain edema.	[26]
	Rat	Endovascular perforation	Tozasertib reduced neuronal apoptosis, and p-JNK attenuated brain edema and improved neurobehavioral deficits.	[27]
Stroke/traumatic brain injury	Mouse	Photothrombosis-induced focal cortical stroke	Knockdown of post-stroke upregulated protein CCR5 inhibited DLK expression.	[28]

**Table 2 cells-13-00333-t002:** DLK inhibitors in (pre)clinical studies.

Tested Inhi- bitor	Clinical Trial or Model Organism (In Vivo/In Vitro)	Experimental Disease Model	Modeled Disease or Disease Mechanism	Effect of DLK Inhibition	Ref
GNE-3511	Mouse	Optic nerve crush MPTP (mitochondrial toxin)	Acute nerve injury Parkinson’s disease (PD)	Reduced c-Jun phosphorylation and neurodegeneration.	[24]
	*D. melanogaster*	*dFmr1* deletion	Fragile-X-syndrome (FXS)	Suppressed defects in neuronal development and behavior.	[29]
	Mouse	Sciatic nerve transection	Nerve injury/neuropathic pain	Reduced mechanical allodynia.	[30]
	Mouse Mouse embryonal DRG explant culture	Cyclophosphamide treatment (p. o.)	Cyclophosphamide-induced cystitis, inflammation, pain, and bladder pathology	Reduced pain and c-Jun phosphorylation in DRG, suppressed histamine-release from mast cells, and neuronal activation in the spinal cord and bladder pathology.	[31]
	Mouse embryonal DRG explant culture	Vincristine NGF withdrawal Axotomy/forskolin Axotomy	Wallerian axon degeneration	Delayed and reduced neurodegeneration. Reduced c-Jun-phosphorylation.	[32,33,34,35,36]
	Mouse sympathetic neurons (with latent HSV infection)	Forskolin, IL-1beta and tetrodotoxin Forskolin + heat shock PI3K inhibition by LY204002	HSV reactivation via neuronal hyperexcitation HSV reactivation by stress	Prevented HSV reactivation.	[37,38,39]
	Mouse cortical and hippocampal neurons	Overexpression of human Tau	Alzheimer’s disease (AD)	Prevented phosphory-lation of JNK + c-Jun and reduced cytotoxicity.	[6]
	Mouse	Deletion of Myelin Regulatory Factor (*Myrf*) (model of demyelination)	Multiple sclerosis (MS)	Blocked c-Jun-phosphorylation and apoptosis of chronically demyelinated neurons.	[40]
GNE-8505	Mouse	SOD1^G93A^ mutation	Amyotrophic lateral sclerosis (ALS)	Reduced p-c-Jun.	[1]
	CMT2 patient-derived iPSC motor neurons	Patient-derived cells in vitro	Charcot-Marie-Tooth neuropathy Type 2 (CMT2)	Reduced p-c-Jun and restored mitochondrial dysfunction.	[41]
IACS-52825	Mouse	Cisplatin treatment (i.p.)	Chemotherapy-induced peripheral neuropathy (CIPN)	Reversed mechanical allodynia.	[42]
IACS-8287	Mouse	Cisplatin treatment (i.p.)	Chemotherapy-induced peripheral neuropathy (CIPN) Chemotherapy-induced cognitive impairment (CICI)	Prevented mechanical allodynia, loss of intra-epidermal nerve fibers, cognitive deficits, and impairments of brain connectivity.	[43]
GDC-0134	Human (Phase I clinical trial)	ALS patients	Amyotrophic lateral sclerosis (ALS)	Terminated due to adverse effects.	[4]
DN-1289	Mouse	Optic nerve crush SOD1^G93A^ mutation	Acute nerve injury Amyotrophic lateral sclerosis (ALS)	Suppressed elevation of p-c-Jun.	[20]
Sunitinib	Human ESC-derived RGCs	Colchicine 1 µM	Axonal injury	Increased dose-dependent survival.	[44]
	Mouse	Impact acceleration (model of TBI)	Traumatic brain injury (TBI)	Increased RGC survival.	[25]
	Mouse Rat	Optic nerve crush	Glaucoma	Administration by eye drop improved RCG survival.	[45]
Tozasertib	Rat Mouse primary RGCs	Laser-induced ocular hypertension Optic nerve transection	Glaucoma, traumatic optic neuropathies	Increased survival of RGCs and reduced axon loss. Increased survival of RGCs	[7]
	Rat	Endovascular perforation	Brain injury after SAH	Reduced apoptosis of neurons.	[27]
	Human ESC-derived RGCs	Colchicine 1 µM	Axonal injury	Increased dose-dependent survival.	[44]
	Mouse primary RGCs	Colchicine 1 µM	Glaucoma	Increased survival of RGCs.	[46]

The reactivation of latent neuronal herpes simplex virus (HSV) infections can cause, besides blisters and sores, severe encephalitis, and it is triggered by neuronal stress. By inducing the initial wave of the increased expression of lytic genes, DLK mediates the reactivation of latent HSV infection in different model systems [37,39,47]. In addition, an association of DLK with different kinds of cancer was found, whereby DLK mRNA expression was positively correlated with lung adenocarcinoma, pancreatic duct adenocarcinoma, sarcoma, and thymoma. The survival rates in kidney renal clear cell carcinoma, kidney renal papillary cell carcinoma, pheochromocytoma, paraganglioma, stomach adenocarcinoma, and uterine corpus endometrial carcinoma were all negatively associated with DLK mRNA expression [48]. In prostate cancer cells, DLK was shown to regulate proliferation and invasion [49]. Thus, the inhibition of DLK is supposed to be a promising drug target for the treatment of several neurodegenerative diseases and, possibly, diabetes. Indeed, some DLK inhibitors have been developed and already tested in vitro and in vivo, and sunitinib and tozasertib have been re-purposed as DLK inhibitors but are in fact multi-kinase inhibitors (Table 2). However, in a phase-1 with patients suffering from ALS treated with GDC-0134, an unexpected increase in a putative ALS biomarker was observed, and no tolerable dose could be found [4]. The development of the DLK inhibitor IACS-52825 was stopped due to dose-independent reversible optic nerve swelling in monkeys [50]. 

DLK belongs to the class of the mixed lineage kinases (MLK), characterized by sequence homology to both serine/threonine and tyrosine kinases in their primary structure, but functioning as serine/threonine kinase [47,48,49]. The class of the MLK can be subdivided into three subclasses: The largest class consists of MLK1-4 (MAP3K9-11 and 21, respectively), sharing 75% sequence identity in their kinases domains and displaying an amino-terminal SH3 domain, followed by the kinase domain, a leucine zipper, the Cdc42/Rac Interacting Binding (CRIB) motif, and a large C-terminal region. Another subgroup consists of the leucine zipper and sterile-alpha motifs (SAM) ZAK or MLK7 (MAP3K20), containing the dimerizing SAM in addition to the leucine zipper domain. DLK forms another MLK subgroup with the leucine zipper kinase (LZK; MAP3K13). The kinases share 90% amino acid sequence identity within their enzymatic and dual leucine zipper domain (Figure 1). 

DLK and LZK are highly conserved orthologues of Wallenda/DLK in *Drosophila melanogaster* and DLK-1 in *Caenorhabditis elegans*, suggesting an important role for DLK in evolution and development. Indeed, while mice lacking *Mlk1*, *Mlk2*, *Mlk3*, or *Lzk* are viable [48,50,51], mice lacking *Dlk* die perinatally and show signs of impaired neuronal development [17]. In line with this finding, the genome aggregation database (gnomAD v 2.1), a repository of data on human genes and their single-nucleotide variants (SNV), calculated for the *MAP3K12* (gene) 41.3 loss of function (LoF) SNV but observed none of them in their different cohorts. Thus, in species as diverse as mice and humans, intact DLK appears to be essential for prenatal development. Yet, the conditional deletion of *Dlk* in mice aged 10 to 12 weeks did not result in gross phenotypic changes, suggesting that in adults, the absence of DLK or its function does not interfere with vital functions under normal conditions [18]. The inhibition of DLK has been proposed as a promising drug target to treat neurodegenerative diseases like amyotrophic lateral sclerosis, Alzheimer’s disease, Parkinson’s disease [1,2,4], glaucoma [7], and diabetes mellitus type 2 [8,9,10]. This suggests that abnormal DLK activity in adults contributes to pathological signaling in various tissues. However, DLK signaling is required for the induction of the pro-regenerative transcriptional program in peripheral nerves after injury [52,53], and it has been shown to be constitutively active in the adult mouse brain, exerting both homeostatic and stress-induced functions [3]. These findings suggest that DLK indeed acts as a “double-edged sword” [54]. Acting as a MAP3K, DLK mainly activates the MAP2Ks MKK4 and MKK7, leading to the phosphorylation and activation of the MAPK c-Jun N-terminal kinase (JNK) [55]. In a refined model, DLK under basal conditions is tethered to the scaffold protein JNK-interacting protein/Islet Brain 1 (JIP/IB1), rendering the kinase inactive. Upon the phosphorylation of JIP/IB1 by JNK on Thr103, DLK dissociates from the scaffold, homodimerizes, autophosphorylates in trans, and becomes active [56,57,58,59]. This model implies that activation of JNK leads to the activation of DLK, which, in turn, stimulates JNK activity, thereby amplifying possibly cell-toxic signals. In addition to activating JNK and being (indirectly) activated by this kinase, DLK also activates other MAPKs [2]. Furthermore, different post-translational modifications that might result in DLK’s proteasomal degradation or enhanced protein stability, protein–protein interactions, subcellular localization, and microRNAs contribute to DLK activity. To better understand DLK function and the regulation of its activity, this review summarizes the regulation of DLK activity via various mechanisms at transcriptional, translational, and post-translational levels.

## 2. Regulation of DLK

### 2.1. Regulation of DLK at the Transcriptional Level

Not much is known about the regulation of the DLK gene expression: The TATA-box—the less core promoter regions of human and mouse DLK gene upstream of exon 1 share 88% identity with completely conserved xenobiotic responsive element-like sites, GC-boxes, and exon 1 in between both species. Using electrophoretic mobility shift assays (EMSA) and reporter gene assays with 5′-deleted promoter fragments, the transcription Sp3 factor was shown to bind to and activate the core promoter in the human neuroblastoma cell line SH-SY5Y [60]. In the 3T3-L1 cell line, the ligand of the peroxisome-proliferator-activated receptor γ (PPARγ) rosiglitazone increased DLK expression, whereas the inhibition of PPARγ either by small hairpin RNA or the receptor antagonist GW9662 suppressed DLK protein and mRNA expression. Two binding-sites for PPARγ and its heterodimer retinoic X receptor were identified via EMSA and chromatin immunoprecipitation assays [61]. In human adipose stromal/stem cells, precursors of mature adipocytes, and bisphenol A increased *DLK* expression, presumably after binding to estrogen receptors [62], but it was not investigated whether the *DLK* promoter contained an estrogen receptor responsive element. Thus, DLK gene expression is regulated by Sp3, the nuclear receptors PPARγ, and the estrogen receptor. Of note, DLK itself has been shown to increase PPARγ gene expression in 3T3-L1 and adipose stromal/stem cells [62,63]. Finally, reduced *Dlk* expression was observed in the lenses of mice deficient in the transcription factors *Mafg* and *Mafk* [64], suggesting that these transcription factors may be involved in *Dlk* expression. 

### 2.2. Regulation of DLK at the Post-Transcriptional Level

The human MAP3K12 gene spans 21.871 nucleotides (nt) and is located on the complementary strand of chromosome 12, GRCh38.p14 Primary Assembly. It is flanked by the gene encoding for the TARBP2 subunit of the RISC loading complex (TARBP2) and the poly(rC) binding protein 2 (PCBP2) gene, both located on the positive strand in opposite directions relative to the DLK gene. The transcripts of the human DLK Isoforms 1 (NM_001193511.2) and 2 (NM_006301.4) differ in a 99 nt stretch that is absent in Isoform 2, resulting in a slightly shorter protein of 859 amino acids (aa) and a calculated molecular weight of 93.2 kDa, instead of 892 aa and 96.3 kDa (NCBI, https://www.ncbi.nlm.nih.gov/gene/7786, 30 June 2023, 14:33. conserved) (Figure 2). 

In various additional model organisms, different isoforms of DLK are expressed, probably due to differential splicing (Table 3). Of note, only Isoform D of Wallenda differs at the protein level from isoforms A, B, and C. 

It is not known how differential splicing of DLK RNA is regulated and whether the isoforms or aberrant splice isoforms exert distinct functions or are differentially expressed in tissues or diseases. An exception is the *C. elegans* DLK-1 gene [65]. DLK-1S (short isoform of DLK-1) interacts with DLK-1L (long isoform), restraining DLK-1L activity. An increase in calcium induces the homodimerization of DLK-1L and its activation, whereby a hexapeptide within the C-terminus of DLK-1L is essential for DLK-1L activity. However, the mammalian homolog of DLK-1 is MAP3K13 (LZK) [65], and although MAP3K13 and MAP3K12 share a high sequence identity in their catalytic and leucine zipper domains (Figure 1), this hexapeptide is not conserved in MAP3K12, making the same kind of regulation described for DLK-1 unlikely to occur for DLK. 

MicroRNAs (miRNAs) provide another mechanism for regulating gene expression at the post-transcriptional level. MiRNAs are small non-coding RNAs of about 22 nucleotides. By directing the RNA-induced silencing complex (RISC) to specific target mRNAs, miRNA can repress target genes and affect various biological responses [66]. In turn, the expression of miRNAs is regulated by several physiological and pathophysiological conditions. *DLK* mRNA is predicted to be a target for miRNAs (TargetScan v8.0; targetscan.org) [67], and several interactions have been validated experimentally (miRTarBase) [68]. In neuroblasts, the downregulation of the entire miR-17 family during neuronal differentiation and upregulation of *DLK* mRNA was observed, and the overexpression of miR-17 and miR-20a reduced *DLK* mRNA [69]. In endothelial progenitor cells (EPC) from type 2 diabetic patients, the expression of miRNA-130a was reduced, and increased *DLK* expression was observed compared to EPC from healthy controls. In line with this finding, the overexpression of miRNA-130a decreased DLK protein expression in EPC, suggesting that DLK expression is regulated by miRNA-130a [70]. The regulation of DLK via miRNA was also demonstrated in a mouse model of Alzheimer’s disease (AD). MicroRNA-191-5p was shown to target the 3ʹ-untranslated region of *MAP3K12*, downregulating DLK expression and alleviating microglial cell injury in the AD mouse model [22]. Yu et al. found, in various prostate cancer cell lines, that the tumor suppressor miR-150-5p downregulates MAP3K12 [71]. These studies show that DLK expression is subject to miRNA regulation. Of note, miRNAs commonly target mRNAs of several proteins, so it is to be expected that a given miRNA downregulates more proteins than DLK only. 

### 2.3. Regulation of DLK at the Post-Translational Level

#### 2.3.1. Phosphorylation of DLK

Already, the first studies on DLK showed that this kinase is heavily phosphorylated, and the overexpression of DLK alone is sufficient to activate this kinase [47,72]. At least some phosphorylation sites and DLK phosphorylating kinases have been identified in the meantime. Upon homodimerization via its leucine zipper domains, DLK becomes auto-phosphorylated in trans at Ser-302 (Figure 3B) [11,56,73]. 

Indeed, the phosphorylation occurring at this residue is crucial for DLK activation, since the mutation of Ser-302 renders the kinase inactive [11,73]. Hence, phosphorylation at this residue is a prerequisite for DLK activation and can serve as a marker for DLK activity [10,11,74]. In addition to DLK, protein kinase A (PKA) phosphorylates DLK at Ser-302 and activates the kinase, linking DLK to the evolutionary conserved mechanism of cyclic AMP-induced axonal regeneration in two mammals, *D. melanogaster* and *C. elegans* [74,75] (Figure 3B). Furthermore, DLK is a substrate for its non-downstream kinase JNK: using stable isotope labeling with amino acids in the HEK 293T cell line (SILAC), followed by mass spectrometry analysis, Huntwork-Rodriguez et al. (2013) identified Thr-43 and Ser-533 as residues that are phosphorylated by JNK (Figure 3B). These findings were confirmed in a murine model of neuronal stress. Phosphorylations at these residues prevent the interaction of DLK with the E3 ubiquitin ligase Pam/Highwire/RPM-1 (PHR), thus stabilizing DLK (Figure 3A) [73]. In an attempt to identify kinases that activate neurodegenerative DLK/JNK signaling in neurons, the inhibition of the MAP4K subfamily of germinal center kinase-IV (GCK-IV), MAP4K4, mis-shapen-like kinase 1 (MINK, MAP4K6), and Traf2- and Nck-interacting kinases (TNIK, MAP4K7) reduced DLK activation, phosphorylation on Thr-43, and protein stability upon nerve growth factor (NGF) withdrawal in murine dorsal root ganglia (Figure 3B). However, the combined knock-down of all three MAP4Ks was needed to protect against NGF-withdrawal-induced DLK/JNK signaling [33]. Using the optic nerve crush model, inhibitors of the GCK-IV kinase family enhanced the survival of retinal ganglia cells but, in contrast to DLK inhibition, did not interfere with axon regeneration [76]. Notably, the overexpression of MAP4K3 (hematopoietic progenitor kinase 1, HPK1), shown to activate JNK signaling [77], phosphorylated MLK3 on Ser-281, corresponding to Ser-302 in DLK [78]. Thus, at least some MAP4K might activate DLK, but not all functions of DLK overlap with those of MAP4K, suggesting additional upstream signals for DLK or an additional regulation of DLK action via other mechanisms, such as (tissue-dependent) dephosphorylation, (tissue-dependent) interaction with various scaffolds or other proteins, palmitoylation, or changing DLK subcellular localization [3,5,9,10,72,73,79,80,81,82]. The study of Daviau et al. (2009) indicates that DLK might also be activated via tyrosine phosphorylation and, thus, involved in a separate pathway. Various experiments showed that platelet-derived growth factor (PDGF) induced tyrosine phosphorylation and subsequent activation of DLK, which was dependent on the cytosolic tyrosine kinase Src. The PDGF-dependent phosphorylation and activation of ERK and Akt was abolished via the RNA silencing of DLK and rescued via the re-introduction of recombinant wild-type DLK, suggesting that PDGF signal propagation depends on DLK. However, the tyrosine residue within DLK phosphorylated by PDGF-induced signaling was not identified [83]. So far, the phosphorylation of DLK that has been described involved activating phosphorylation. In mouse embryonic stem cells, two Akt phosphorylation sites within DLK, Ser584 and Thr659 in murine DLK, were identified. The Akt-induced phosphorylation of these residues reduced DLK kinase activity, whereas the overexpression of these DLK mutants rendered the kinase more active, suppressing the self-renewal of mouse embryonic stem cells [84]. 

#### 2.3.2. Dephosphorylation of DLK

Kinase–phosphatase interactions are a well-known component of cellular responses and signaling pathways, affecting kinase phosphorylation, expression levels, and interactions with other proteins. In invertebrates, DLK activity has been shown to be regulated by the protein phosphatases Mg^2+^/Mn^2+^ dependent (PPM)-1 and PPM-2, as well as the protein phosphatase 2A (PP2A) [5,85,86]. In mammals, the inhibition of protein phosphatases 2A [5,72,83] and 2B [10,12,72,87] affected DLK activity. 

Several studies described the roles of the PHR proteins as key regulators of presynaptic differentiation and function, thereby fundamentally affecting neuronal development [88,89,90]. Among the PHR proteins, the Regulator of Presynaptic Morphology (RPM)-1 has been shown to negatively regulate DLK-1 as part of an ubiquitin ligase complex in *C. elegans* [91,92]. In 2011, Tulgren et al. provided additional evidence that PPM-1, a serine/threonine phosphatase homologous to human PPM1A, acts as a second negative regulatory mechanism downstream of RPM-1 to control the DLK-1 pathway in *C. elegans* [85]. However, the involvement of PPM-1 was shown to act at the level of PMK-3 (p38 MAPK) and not directly at the level of DLK-1 (MAPKKK), as previously described by Takekawa et al. in mammalian cells [93]. In contrast, the serine/threonine Protein Phosphatase Magnesium/Manganese-dependent 2 (PPM-2) has been described in transgenic animals, genetic, and biochemical approaches for directly regulating DLK-1 [86]. Baker et al. (2014) demonstrated that PPM-2 acts on DLK-1 at the Ser-874, regulating its phosphorylation and activation. However, the authors note that the activation of RPM-1 is a prerequisite for the activity of PPM-2 on DLK-1, and this PHR protein employs ubiquitination and phosphatase-based mechanisms to inhibit DLK-1. This observation is based on immunoprecipitation approaches, followed by mass spectrometry, immunoblot, and immunofluorescence analysis, in which Baker et al. showed that RPM-1 binds to and positively regulates PPM-2. As RPM-1 has previously been shown to be a part of a neuronal complex involving multiple proteins [91,92], more precise approaches are lacking to confirm that RPM-1 directly binds to PPM-2. In addition, the small segment of the *C. elegans* DLK-1 containing Ser-874 is not conserved with mammalian DLK, and no functional orthologue of PPM-2 is known in vertebrates (Yan et al., 2012 [65]).

In *D. melanogaster*, Valakh et al. (2013; 2015) showed that cytoskeletal dysregulation activates Wallenda/DLK [94,95]. Hayne and DiAntonio (2022) hypothesized that the disruption of the cytoskeletal structure is mediated by the inhibition of the serine/threonine protein phosphatase 2A (PP2A), which ultimately triggers DLK activation [5]. In addition to cytoskeletal perturbations, the dysregulation of PP2A function leads to a variety of cell type-specific cellular dysfunctions, including defects in mitotic processes, and cell death [96]. In mammalian cells, pharmaceutical intervention with okadaic acid, an inhibitor of serine/threonine phosphatases including PP2A, showed an increase in the abundance of phosphorylated DLK [72], as well as in DLK activity [83]. However, these experiments have not definitively shown whether DLK is a direct target of PP2A. Daviau et al. (2009) also showed that the protein phosphotyrosyl-phosphatase inhibitor vanadate induced an approximately 3.5-fold increase in DLK activation compared to activity without treatment and an approximately 1.5-fold greater effect than the effect of okadaic acid. In contrast to PP2A, the authors state that the effect of vanadate on DLK is not a direct effect on the kinase but is mediated through a Src kinase-dependent pathway, and they showed similar results for induction of the PDGF receptor signaling pathway [47,83]. 

The calcium/calmodulin-dependent serine/threonine protein phosphatase 2B (calcineurin) has also been implicated in the regulation of DLK activity in several studies [10,12,72,87]. Mata et al. (1996) showed in rat aggregating glial cell cultures that the inhibition of calcineurin by its selective inhibitor cyclosporin A (CsA) affected DLK electrophoretic mobility only after membrane depolarization by veratridine and not in unstimulated conditions, highlighting membrane depolarization as a prerequisite for the effect of CsA action on DLK [72]. Consistent with this, Daviau et al. observed no effect of CsA on DLK activity under unstimulated conditions in COS-7 cells transfected with a vector encoding a wild-type T7-DLK sequence [83]. Controversially, other studies in insulin-producing beta-cells have observed an effect of CsA on DLK activity: (i) the enhancement of DLK-dependent c-Jun phosphorylation [87], (ii) increase in DLK kinase activity by an in vitro assay using casein as a substrate and the induction of beta-cell apoptosis [12], (iii) augmented DLK-dependent phosphorylation of the c-Jun N-terminal kinase (JNK), increased phosphorylation and activation of DLK at Ser-302, and increased nuclear translocation of DLK with concomitant increase in beta-cell apoptosis [10]. Most of these DLK/calcineurin-dependent effects were also observed after calcineurin inhibition by tacrolimus (FK506), another structurally distinct, selective calcineurin inhibitor. It should be noted that Mata et al. (1996) [72] and Daviau et al. (2009) [83] used higher concentrations of CsA (30 µM) and a longer treatment time (4 h) than the other studies (5 or 10 µM for 10 or 30 min). Although there is considerable evidence that DLK activity is partly regulated by the serine/protein phosphatase 2B, it is not known whether DLK and calcineurin interact directly. Duque Escobar et al. (2021) revealed a distinct φLxVP motif (aa 362–365) within DLK for interacting with the calcineurin A and B subunit interface, as already described for the NFAT transcription factor [10,97]. The authors used several mutations to show that Val-364 prevented protein–protein interactions and exhibited increased DLK activity, measured as the phosphorylation of the downstream JNK, inhibition of CRE-dependent gene transcription, and induction of beta-cell apoptosis [10]. Furthermore, the activation of DLK by the calcineurin inhibitors might contribute to the pathogenesis of post-transplant diabetes mellitus observed after treatment with these immunosuppressant drugs [98]. 

#### 2.3.3. Palmitoylation of DLK

The dual leucine zipper kinase is crucial for retrograde signaling after injury in neurons and important in brain development, but the activation of DLK can also lead to apoptosis and neuronal degeneration in different disease models, such as ALS or Alzheimer’s disease. Due to the importance of the DLK for cell fate, the activity of DLK, at least in neurons, needs to be highly restricted in temporal and spatial manners and confined to local events. In a search for an explanation on how a bioinformatically predicted soluble protein could travel from distant axonal regions back to the nucleus in a controlled way, Holland et al. found and validated an evolutionary conserved cysteine residue (C127, in human DLK) in the DLK as a site for post-translational modification via palmitoylation (Figure 4) [82].

Palmitoylation is a fast, dynamic post-translational modification that is catalyzed by palmitoyl acyl transferases such as zinc aspartate-histidine-histidine-cysteine (zDHHC) family members and reversed by acyl protein thioesterases. Palmitoylated proteins are tethered to membranous structures like vesicles, the golgi, the plasma membrane, or the outer mitochondrial membrane [99]. Holland et al. (2016) showed that palmitoylated DLK is targeted to motile trafficking vesicles, allowing it to traffic retrogradely in the event of axonal injury [82]. In addition, palmitoylation is essential for the interaction of DLK with the scaffold protein JNK-interacting protein-3 (JIP3), resulting in the phosphorylation of DLK’s direct and indirect substrates, like MKK4/7, JNK3, and c-Jun, rendering palmitoylation as a novel mechanism for regulating DLK activity [100]. DLK palmitoylation does not require the phosphorylation or homodimerization of DLK; rather, it brings DLK and its substrates into close proximity [82,101]. A model proposed that palmitoylation tethers DLK and JNK3 to the same axon vesicle membranes and leads to a forward loop, whereby DLK, through activation of MKK4/7, phosphorylates JNK3, which, in turn, activates DLK, thus perpetuating DLK signaling [100]. This model implies that JNK contributes to the activation of DLK, which has been shown before in non-neuronal and neuronal cells [11,57,73]. DLK subcellular localization is also palmitoylation-dependent in non-neuronal cells (HEK293T), suggesting that inhibitors of palmitoylation might inhibit DLK activation [102]. Although not all MLKs are palmitoylated [100], many more proteins besides DLK undergo this post-translational modification; thus, interfering with palmitoylation is expected to be very unselective. In an optic nerve crush (ONC) model, Niu et al. (2020) identified the palmitoyltransferase ZDHHC17 as a DLK palmitoylating enzyme [103]. Taken together, the palmitoylation of DLK represents a mechanism to bring DLK and its substrates into proximity and direct the subcellular localization of this kinase. Stress signals are reported to increase the palmitoylation of DLK, but it remains unknown how palmitoyl acyl transferases become activated. 

#### 2.3.4. Regulation via Protein–Protein Interactions

DLK protein content and, therefore, activity is regulated via the interaction with diverse proteins. The Regulator of Presynaptic Morphology 1 (RPM-1) in *C. elegans* and Highwire (Hiw) in *D. melanogaster* were the first proteins demonstrated to interact with DLK-1 and Wallenda, respectively. Together with the human protein associated with Myc (PAM), these proteins are termed PHR (PAM, Highwire, RPM-1) proteins, which are huge proteins with more than 3700 aa containing diverse enzymatic activities and mainly regulate synapse formation and axon termination [104]. In *C. elegans*, *D. melanogaster*, and the dorsal root ganglia of mammals, RPM-1, Highwire, and PHR, respectively, ubiquitinate DLK-1/Wallenda/DLK, leading to the kinase’s proteasomal degradation, thus terminating kinase activity [104]. However, in mammals, PHR does function independently of DLK in some neuronal contexts [104]. Using primary dorsal root ganglia cells as a model, Huntwork-Rodriguez et al. (2013) showed that the interaction between DLK and the E3 ubiquitin ligase PHR is regulated via the JNK-induced phosphorylation of DLK: the phosphorylation of DLK on Thr-43 and Ser-533 stabilizes DLK, presumably preventing the interaction with PHR1. Under basal conditions, when DLK is not phosphorylated, its abundance is regulated by a balance between PHR1 and the deubiquitinase ubiquitin-specific peptidase 9, X-linked (USP9X) [73]. In addition to PHR, the FK506-binding protein-like (FKBPL) and the FK506-binding protein 8 (FKBP8) were identified as DLK-interacting proteins [81]. By interacting with the N-terminus of DLK, containing its kinase domain, the N-terminus of FKBPL, containing its peptidyl-prolyl isomerase domain, inhibited DLK activity and reduced its protein stability. Both FKBPL and FKBP8 induced DLK degradation via the lysosomal pathway. Additionally, FKBP8-mediated DLK degradation was prevented when Lys-271 was mutated to Arg, which was shown to function as an ubiquitination and SUMOylation site. Thus, FKBP8 induced DLK lysosomal and proteasomal degradation [81]. 

Whereas the interactions of DLK with PHR, FKBP8, and FKBPL result in the degradation of the kinase and, thus, in its inactivation, the interaction with the scaffold protein JIP brings DLK and its substrates in proximity, thereby increasing and perpetuating DLK activity [58,59]. In a thorough mutational analysis, Mooney and Whitmarsh (2004) identified two JIP1-interaction sites within DLK, whereby the mutations of Phe-117, Leu-397, and Asp-398 to alanine severely impaired the DLK–JIP1 interaction and JNK phosphorylation [105]. The DLK–JIP–JNK interaction is regulated via the phosphorylation of JIP: the phosphorylation of JIP1 on Thr-103 by JNK induces the dissociation of DLK from the scaffold protein, DLK homodimerization, activation, and, ultimately, the phosphorylation of JNK. In murine brain cell lysates, the phosphorylation of JIP1 on tyrosine residues by Src kinases increased the affinity of DLK to JIP1, thereby strengthening this interaction and preventing DLK activation [57,58]. Of note, palmitoylation contributes to bringing DLK, JIP, and JNK into proximity [100]. The interaction of DLK with specific JIPs seems to be tissue-dependent and contributes to selective DLK action. In the cerebellum of adult mice, where JIP1 was preferentially expressed, DLK was constitutively active in the absence of injury signals. In the murine forebrain, where JIP3 and another scaffold protein of DLK, Plenty of SH (POSH), are expressed, DLK becomes activated only by neuronal injury [3,106]. Further analysis revealed that in the cerebellum but not in the forebrain after neuronal injury, DLK regulated insulin growth factor 1 signaling. Hence, the regulation of DLK action seems to depend on the interaction with particular scaffold proteins and on the tissue [3]. 

The heat-shock proteins (HSP) are another group of proteins regulating DLK protein abundance and, therefore, enzymatic activity. The HSP90 acts as a chaperone, which, in contrast to other HSPs, facilitates the maturation, complex assembly, localization, and ligand binding of signal transduction proteins like kinases and nuclear receptors [79,107]. Using HSP90 and DLK inhibition and co-immunoprecipitation assays, an interaction between DLK and HSP90 was shown to occur in *D. melanogaster*, embryonic DRG neurons and the HEK cell line, whereby the inhibition or loss of HSP90 (or its fly ortholog Hsp83) reduced DLK protein content [79]. Hence, the interaction between HSP90 and DLK preserves DLK, possibly by preventing the interaction between DLK and PHR1 leading to DLK proteasomal degradation. In contrast, in the cell line COS7, the interaction of activated DLK with HSP70 results in DLK proteasomal degradation, which was dependent on the HSP70 co-chaperone CHIP (C-terminus of HSP70 interacting protein), an E3 ubiquitin ligase [80]. Notably, both chaperones, HSP90 and HSP70, can interact with the co-chaperone CHIP and induce the proteasomal degradation of their respective clients [108]. This suggests that the outcome of the interaction of DLK with HSP chaperones does depend on the expression of the co-chaperone CHIP. 

In addition, a direct interaction between DLK and calcineurin has been demonstrated [10].

#### 2.3.5. Regulation of DLK via Its Oligomerization

The homodimerization of DLK via its leucine zipper has been shown to be essential for DLK activity [56]. Experiments conducted in different cell lines (NIH3T3, COS-1) indicated that the formation of high-molecular DLK polymers occurred in response to the apoptosis-inducing agent Calphostin C, independent of its ability to inhibit protein kinase C (PKC) [109,110,111]. DLK oligomerization was abolished by the tissue-transglutaminase (tTG) inhibitor monodansylcadaverine, indicating that a tTG-catalyzed protein crosslinking reaction was the underlying cause. A model emerged whereby the Calphostin-C-induced intracellular rise of Ca^2+^ stimulates Ca^2+^-dependent tTG2 crosslinking-activity, leading to an increase in the DLK polymers. Oligomerization then increases DLK activity and, hence, activates the JNK-Pathway. After JNK-activation, the apoptosis regulator Bax translocates to mitochondria and induces caspase activation, ultimately leading to apoptosis [109,110,111]. It remains unknown which domains of DLK mediate the oligomerization. Nevertheless, bringing single DLK molecules together, either as homo- or oligomers, leads to increased DLK activity. 

## 3. Conclusions

DLK protein levels, enzymatic activity, and localization are tightly regulated at different levels and via distinct, possibly tissue-dependent mechanisms. In addition, DLK activity and proteins levels are subject to further fine tuning via interactions with several other proteins and/or via post-translational modifications. Furthermore, transcriptional and post-transcriptional mechanisms contribute to DLK regulation. The regulation of DLK by miRNA or siRNA might be a possible tool to modulate DLK expression in a tissue-selective manner. Further elucidation of these diverse mechanisms will contribute to the identification of drug targets to selectively regulate DLK action. Thus, preventing the interaction with cell-selective scaffold proteins [3] by small molecule protein interaction inhibitors or interfering with subcellular localization of DLK [9,102] might inhibit DLK action more selectively than ATP-competitive small-molecule inhibitors. In addition, other than that issue, the dual leucine zipper mediates DLK homodimerization required for kinase activity, though not much is known about its impact regulating DLK activity. Do proteins, besides the CREB-regulated transcriptional coactivators (CRTCs), interact with the dimerized leucine zipper of DLK [112] and, possibly, interfere with the kinase’s action? Does the dimerization induce a conformational change in DLK, enabling the access to the catalytic domain, thereby regulating DLK at yet another level? The answers to these questions might open an avenue to the cell- or tissue-selective regulation of DLK action. 

## Figures and Tables

**Figure 1 cells-13-00333-f001:**
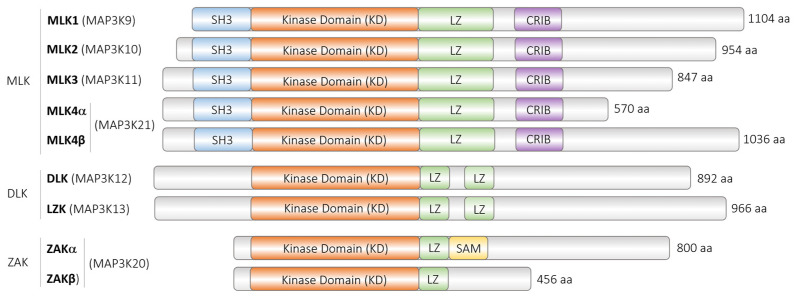
Mixed-lineage kinase (MLK) subfamilies. Schematic depiction of the functional domains of different MLK family members (not to scale). MLK (mixed-lineage kinase), DLK (dual leucine zipper kinase), LZK (leucine zipper kinase), ZAK (sterile alpha motif and leucine zipper containing kinase AZK), SH3 (Src homology 3 domain), LZ (Leucine Zipper), CRIB (Cdc42/Rac interactive-binding), and SAM (sterile-α motif). NCBI RefSeq accession numbers (if applicable): MLK1/MAP3K9 (NP_001271159.1, Isoform 2), MLK2/MAP3K10 (NP_002437.2), MLK3/MAP3K11 (NP_002410.1), MLK 4/MAP3K21 (CAC84639.1, Isoform 1/α; NP_115811.2, Isoform 2/β), DLK/MAP3K12 (NP_001180440.1, Isoform 1), LZK/MAP3K13 (NP_004712.1, Isoform 1), and ZAK/MAP3K20 (NP_057737.2, Isoform 1/α; NP_598407.1, Isoform 2/β).

**Figure 2 cells-13-00333-f002:**
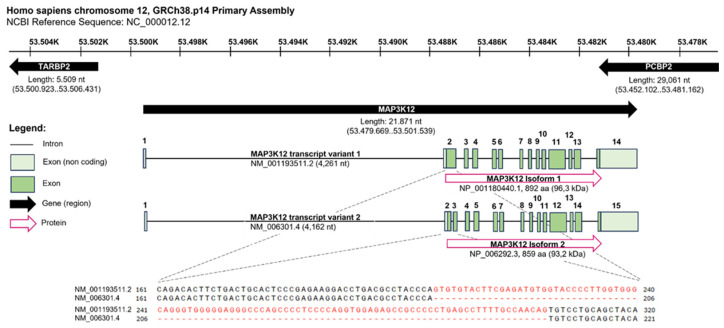
The human MAP3K12 gene spans 21.871 nucleotides (nt) and is located on the complementary strand of chromosome 12, GRCh38.p14 Primary Assembly. It is flanked by the gene encoding for the TARBP2 subunit of RISC loading complex (TARBP2) and the poly(rC) binding protein 2 (PCBP2) gene, both located on the positive strand in opposite directions relative to the DLK gene. The transcripts of the human DLK Isoforms 1 (NM_001193511.2) and 2 (NM_006301.4) differ in a 99 nt stretch absent in Isoform 2, resulting in a slightly shorter protein of 859 amino acids (aa) and a calculated molecular weight of 93.2 kDa instead of 892 aa and 96.3 kDa. Data were retrieved from NCBI (https://www.ncbi.nlm.nih.gov/gene/7786, 30 June 2023, 14:33). The figure created using BioRender.

**Figure 3 cells-13-00333-f003:**
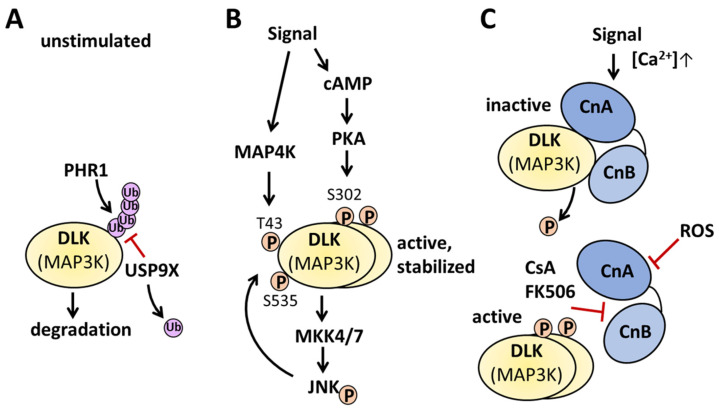
Examples for regulation of DLK activity. (**A**) Under basal unstimulated conditions, unphosphorylated DLK protein abundance is regulated by the E3 ubiquitin ligase PHR1 and the deubiquitinase USP9X. (**B**) Signals activating cAMP and PKA phosphorylate dimerized DLK on Ser-302, leading to the activation of MKK4/7 and JNK. JNK, in turn, phosphorylates DLK on Thr-43 and Ser-535, preventing the interaction with PHR1, thereby stabilizing DLK. Other signals activating MAP4K phosphorylate DLK on Thr-43 and stabilize DLK. (**C**) Upon an increase in the intracellular calcium concentration, calcineurin interacts with monomeric or dimeric DLK and dephosphorylates the kinase. The inhibition of calcineurin by ROS prevents the dephosphorylation of DLK, whereas the interaction of immunophilin-bound CsA or FK506 displaces DLK from the calcineurin interaction site, and DLK dimerizes and autophosphorylates in trans. For further information, please see the text.

**Figure 4 cells-13-00333-f004:**
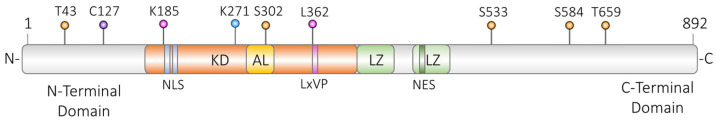
DLK amino acid residues are modulated at the post-translational level. Isoform 1 of human DLK is depicted; T—threonin; C—cystein; K—lysin; S—serin; orange dots—phosphorylation and, possibly, dephosphorylation sites; violet dot—palmitoylation site; blue dot—ubiquitylation and SUMOylation site. KD—kinase domain; AL—activation loop within the kinase domain; LZ—leucine zipper for homodimerization; NLS—bipartite nuclear localization site; NES—nuclear export site; K185—ATP binding site; L—leucine; x—any amino acid; V—valine; P—proline; LxVP—interaction site with calcineurin. For further information, please see the text.

**Table 3 cells-13-00333-t003:** MAP3K12 Isoforms in different model organisms.

Model Organism	Protein	Accession Number	Isoforms	Length
Homo sapiens	MAP3K12, ZPK	NP_001180440.1	Isoform 1	892 aa
Homo sapiens	MAP3K12, ZPK	NP_006292.3	Isoform 2	859 aa
Mus musculus	MAP3K12, DLK	NP_001157115.1	(not described)	888 aa
Rattus norvegicus	MAP3K12, MUK	NP_037187.1	(not described)	888 aa
Mesocricetus auratus	MAP3K12, DLK	XP_040609646.1	Isoform X1	920 aa
Mesocricetus auratus	MAP3K12, DLK	XP_012966775.1	Isoform X2	862 aa
Mesocricetus auratus	MAP3K12, DLK	XP_005067383.1	Isoform X3	892 aa
Caenorhabditis elegans	DLK-1	NP_001021443.1	long	928 aa
Caenorhabditis elegans	DLK-1	NP_001021445.1	short	577 aa
Drosophila melanogaster	Wallenda, WND	NP_649137.3	Isoform (A)	977 aa
Drosophila melanogaster	Wallenda, WND	NP_788540.1	Isoform (B)	977 aa
Drosophila melanogaster	Wallenda, WND	NP_788541.1	Isoform (C)	977 aa
Drosophila melanogaster	Wallenda, WND	NP_001189132.1	Isoform (D)	950 aa

## Data Availability

Not applicable.

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
