# Peer review of "Regulation of the Activity of the Dual Leucine Zipper Kinase by Distinct Mechanisms"

_cells, 2024, doi:10.3390/cells13040333_

Round 1
Reviewer 1 Report
Comments and Suggestions for Authors
In this manuscript, the authors reviewed the different cellular mechanisms that control the activity of DLK, a kinase that is obviously important for development.
The review is clear and well written. The figures helped well to understand the regulations explained in the text.
I have a few minor comments
1.A “Introduction” and a “result” part in a review paper is rather unusual…they should be removed and new titles have to be found
2. In the footnote to table 1, it should be added that 3 of the 4 isoforms of Drosophila diverge at the RNA level but not at the protein level.
3. Clarify the last sentence of point 2.2 (lines 151-152)
4. just add a few sentences to remind some of the cellular functions of DLK so that it will be of interest to develop new drugs to regulate its action
Author Response
We thank this reviewer for the helpful comments on our manuscript! To the points in detail:
1 - we now changed "introduction" to "(patho)physiological actions of DLK" and "results" to "Regulation of DLK"
2 - We now added in the text, that only the isoform D of Wallenda differs at the protein level as well (line 160 to 161)
3 - We now clarified the sentence in line 151-152 and sincerely apologize for the bewilderment we caused.
4 - We did add some sentences (line 32 - 61 and table 1) about the function of DLK
Reviewer 2 Report
Comments and Suggestions for Authors
The paper is generally well written.
My specific comments:
1. The introduction Should be more inclusive of diseases e.g Parkinson's diseases, cancer etc. I think DLK is implicated with many other diseases as evident by the recent research.
2. For better understanding of the reader, the "double-edged sword" role can be illustrated showing schematic pathways. (line 72)
3. To my opinion, many injury models and database reporting the potential inhibitors to DLK activation can be inserted to have a comprehensive knowledge for the readers. (line 127)
4. In addition to invertebrates model animal, data for DLK activation/ inhibition regarding vertebrates especially human, should be included. In recent research, data on vertebrates are available. Line 245
5. The 3 D picture of potential inhibitor binding sites in DLK may add more value to this article.
Comments on the Quality of English Language
Satisfactory
Author Response
We sincerely thank the reviewer for the very helpful comments! To the points in detail:
1 - We now described the role of DLK in additional diseases among them cancer (lines 31 - 62) and added a new table 1, listing the role of DLK in several neuronal diseases.
2 - with the term "double-edged sword" we refered to the title of the excellent review by Tedeschi and Bradke (2013): "The DLK signalling pathway - a double edged sword in neuronal development and regeneration", meaning that DLK can mediate pathological and physiological pathways. Unfortunately, pathological and physiological pathways do not always differ much: in postnatal beta-cells, DLK is important for their proliferation, signalling via JNK3 (Tenenbaum et al., 2020), whereas in the beta-cell line HIT DLK induces apoptosis signalling via JNK (with no distinction in the JNK isoform) (Börchers et al., 2017; Wallbach et al., 2016; Duque Escobar et al., 2021). In the ApoE4-dependent induction of APP gene transcription, DLK signals via the MAPK ERK (Huang et al., 2017). Thus, it seems impossible to predict which pathway of DLK signalling is physiological or pathological. Therefore, we refrained from drawing as figure as requested by the reviewer.
3 - we no added a new table 2, summarizing the studies/preclinical and clinical trials in which DLK inhibitors were used.
4 - we included the table 1 showing mainly vertebrate models with effect of DLK on neurological diseases.
5 - the aim of this review is not the comparison of the various DLK inhibitors or their binding-sites within DLK (and possibly LZK). Our aim was to coprehensively show the distinct mechanism of DLK regulation.
Reviewer 3 Report
Comments and Suggestions for Authors
The authors prepared a comprehensive review about the regulation of DLK activity, which is involved in embryonic development and several diseases. It is very well written and was a pleasure to read. I have some comments.
1. Page 4, line 151: What is the definition of “selective proteins” and what does this particular sentence imply for DLK? That it is not a selective protein? Please elaborate if it is an important issue.
2. It is mentioned that splicing isoforms are different among different organisms. Usually, splicing is very context dependent. Are there also splice isoforms known in different tissues and with different activities? Or perhaps aberrant splice isoforms in disease?
3. Why is chapter 2 named “results”, as there is no section on the method of the article search strategy.
4. Overall, the references are relatively old <2020. Please check if you missed some recent articles.
5. Figure 2 is not of very high quality and the lower row is unreadable.
6. The conclusions are rather superficial and merely a repetition of the results. The reader is probably interested in your view of what these results mean for further basic research and the potential for translational research for the diseases that are mentioned in the introduction and throughout the text.
7. In line with the latter remark, it would improve the value of this review when a table could be added listing the drugs that are used/under investigation to target DLK activity and its signaling pathway, and the (pre)clinical results.
Author Response
We sincerely thank this reviewer for his encouragung and helpful comments on our manuscript. To the points in detail:
1 - We apologize for the bewilderment by choosing this term. This sentence is now - hopefully - better understandable (lines 191 - 193).
2 - Unfortunately, no much is known about different splice isoforms of DLK in different tissues or about aberrant splice isoforms in diseases. However, in C-elegans, DLK-1 (which corresponds to mammalian MAP3K13 alias LZK) a long and a short isoform are observed, whereby the interaction of DLK-1L with DLK-1S renders DLK-1L inactive. Calcium increase results in the homodimerization of DLK-1L and its activation, whereby a C-terminal hexapeptide is required (Yan and Ying, 2012). This hexapeptide is not conserved in DLK (MAP3K12). We now describe these findings (line 163 - 172).
3 - We changed "results" to "regulation of DLK"
4 - We now added several more recent paper, mainly describing novel functions of DLK or some DLK inhibitors.
5 - We now placed figure 2 into a different place which will hopefully improve the readibility of the last line to this figure.
6 - We now rewrote the conclusions and pose some - in pur minds - important questions about DLK whose answers might hopefully lead to new drug targets within DLK.
7 - We now added the table 2, describing the effect of diverse DLK inhibitors in preclinical studies/trials and in the one phase-1 clinical trial.
Round 2
Reviewer 3 Report
Comments and Suggestions for Authors
No furhter comments. I like Table 2. It was a lot of work. Probably you have to adjust the lay-out and may want to place this table where you discuss the druggability.